# Application of the Teaching Personal and Social Responsibility Model in the Secondary Education Curriculum: Implications in Psychological and Contextual Variables in Students

**DOI:** 10.3390/ijerph18063047

**Published:** 2021-03-16

**Authors:** David Manzano-Sánchez, Sixto González-Víllora, Alfonso Valero-Valenzuela

**Affiliations:** 1Department of Physical Activity and Sport, Faculty of Sport Sciences, University of Murcia, Santiago de la Ribera, 30720 Murcia, Spain; avalero@um.es; 2Department of Physical Education, Arts Education, and Music, Faculty of Education, University of Castilla-La Mancha, 16001 Cuenca, Spain

**Keywords:** methodology, motivation, pedagogical model, physical education, psychology, school content

## Abstract

The aim was to implement a value-promoting programme (Teaching Personal and Social Responsibility, TPSR) and to assess its impact on psychological and contextual variables in students comparing the differences among one group that applied it in several subjects, another group only in Physical Education (PE), and a control group. Method: The programme was applied for eight months with 257 students from three secondary school centres (151 boys and 106 girls) with a mean age of 15.97 years (SD = 2.31). They were in three groups: one group with 67 students (control group), one group with 90 students receiving at least 60% of the total teaching time using the value-promoting programme (experimental group for global education, EG-GE), and one group with 100 students (experimental group for physical education only, EG-PE). The main improvements in the results were found in the EG-GE for responsibility, psychological mediator index, self-determination index, resilience, climate, and prosocial behaviour. In the EG-PE, improved results were observed in the self-determination index, classroom climate, and prosocial behaviour. Female and EG-GE students improved much more than male and EG-PE students. The outcomes in psychological variables can be higher if TPSR is applied to the whole subjects apart from physical education. These results are even more pronounced for female students in personal and social responsibility. It is worth highlighting the importance of coordinating educational institutions to facilitate the involvement of the greatest number of teachers.

## 1. Introduction

Social demand calls on education to provide students with tools to be able to adapt to the constant changes and requirements of education and a society full of knowledge where autonomous work within a collaborative network is essential. Among the models that promote these values, we can highlight Hellison’s Teaching Personal and Social Responsibility model [1], generally applied to Physical Education (PE) and after-school activity contexts [2]. Don Hellison’s scholarship has had a lasting impact on the academic literature, policy, and practice of physical education and sports pedagogy, both in the past and nowadays [3].

The most current reviews [4,5,6] concluded that this model in PE shows much evidence of its effectiveness in reducing aggressiveness, solving problems, and improving responsibility and motivation. In addition, they suggest that new lines of research should consider the possibility of it being introduced into other school subjects to enhance the results found.

In this regard, studies such as Richards et al. [7] have provided a list of strategies for promoting social and emotional learning with the Teaching Personal and Social Responsibility Model (TPSR), and Gordon et al. [8] expressed their concern for the transfer of learning acquired by students beyond PE and sports. Wright et al. [9] concluded that TPSR could be applied in the secondary school curriculum, and along these lines, Leo et al. [10] showed Continuous Professional Development (CPD) as a possible solution for adapting the model to other subjects.

The first study that considered the use of TPSR in another subject was Escartí et al. [11], with primary school students, where the fidelity of TPSR was sought, concluding that although worse values were obtained in transfer and lifestyles promotion compared to the other results found, they suggest the school curriculum as the ideal framework for the development of TPSR, going beyond the subject of PE alone.

Recent studies have revealed positive results after the implementation of TPSR in schools, not only in variables such as motivation, responsibility, or autonomy, but also in the classroom climate among primary and secondary school students [12]. Furthermore, this methodology is considered suitable for teachers from different subjects and useful for teaching values and improving their lessons, with teamwork between teachers being essential [13]. Therefore, the promotion of responsibility could contribute to the improvement of other factors such as resilience, prosocial behaviour, the perception of violence, and, consequently, classroom climate, factors that have been considered in various studies, as can be seen in the reviews [5,6].

Another important variable to take into account is gender. When the TPSR model was applied, it appeared that the values of personal and social responsibility increased for both boys and girls [14], but the results were higher in boys compared to girls [15]. Following the study by Smith et al. [16], male students generally have higher levels of basic psychological need satisfaction, as well as intrinsic motivation, compared to female students. However, the study by Manzano-Sánchez et al. [17] recently showed the differences relating to TPSR. They found that the TPSR programme can be used to improve responsibility in both, but only female students improved their basic psychological needs and intrinsic motivation.

The present study sought to follow the criteria for applying TPSR [5]: a sample size larger than 15 students, minimum duration of one academic year, applied at least twice a week, with adapted content and a good environment. In addition, as stated by Gray et al. [18], achieving pedagogical changes requires a commitment between student learning, values, and professional needs. Therefore, data are collected, quantitively and qualitatively, from both students and teachers, which, as proposed by Wright et al. [19], is essential to complement the study, compare the results, and find out the implications for students and teachers.

The aim of the present study was to check the effects of the application of TPSR on secondary education in responsibility, motivation, satisfaction of basic psychological needs, resilience, prosocial behaviour, violence, and classroom climate, comparing the differences among a group with several subjects, a group of subjects in a PE class (as has been traditionally done so for this model) and a control group. Secondly, an attempt was made to see the differences depending on gender.

## 2. Materials and Methods

### 2.1. Design

Following Anguera et al. [20], a mixed method approach was applied based on qualitative (continuous development intervention with observational analysis) and quantitative instruments (different tests were applied at the beginning and end of the programme).

The intervention programme lasted for eight months. The content was selected according to the current education laws in Spain [21], and different tasks were designed for each subject to work on all of the levels of responsibility [22] (Table 1). All students answered a questionnaire before and after starting the programme to check the values of the psychological variables. The questionnaire was administered in a quiet environment for 45 min.

### 2.2. Participants

Participants were selected based on accessibility and convenience. The exclusion criteria were: (a) did not complete all test scales, (b) did not complete both tests (pre- and post-test measurements), (c) did not include double answers and did not answer at least 90% of the exercises. The final sample consisted of 257 students from three different centres in Murcia (Spain) with a low–middle socio-economic context. There were 67 in the control group (CG), 90 in the experimental group in global education (EG-GE), and 100 TPSR only in PE (EG-PE). In the EG-GE group, TPSR was applied in at least 60% of the curriculum content (of the 30 h per week of classes that students have in educational centres, the EG-GE group received a minimum amount of 18 h of instruction from their teachers, implementing the TPSR), among others: Maths, Geography, Music, Language and Literature, and Foreign Language (this was due to the coincidence of the different participant teachers, which was between 60% and 80% of all classroom lessons). There were statistically significant differences in age (CG = 17.34, EG-PE = 14.67, and EG-GE = 13.73) and gender (female CG = 44.77%, EG-PE = 63.00%, EG-GE = 64.44%). The groups were composed for accessibility and convenience. The sample was composed of 151 boys and 106 girls with a mean age of 15.28 years (SD = 3.20). In total, 30 teachers participated in the study. There were 11 men and 19 women aged between 27 and 58 years, with between 3 and 30 years of experience. The teachers taught various subjects, specifically Physical Education (4), English as a Second Language (3), French as a Second Language (2), Spanish Language (3), Mathematics (3), Geography and History (2), Technology (2), Music (2), Ethic Values (2), Religion (2), and Physics and Chemistry (2).

### 2.3. Procedure

#### 2.3.1. Teacher-Specific Professional Development

Teacher-specific professional development is needed in order to implement any educational programme [10]. All teachers held their lessons in their usual space in the corresponding classroom, using the gym or the school playground in the case of PE. None of them had previous experience with the intervention, except one PE teacher, with all of them receiving a training course in TPSR as part of a school seminar. In this course, they received a new TPSR training session, similar to but simpler than the initial training session, and at the end of this training session, the teachers could ask questions.

The teachers were trained in TPSR following the suggestion by Pascual et al. [23] and other authors referenced in that research. (1) A course with a complete explanation about the programme: the pillars and strategies of TPSR were teaching with general and specific strategies and a theorical practice lesson. At the end of this seminar, the main researchers gave the teacher different materials to find out more about the different strategies [24]. This course lasted five hours. (2) Continuous training: during the programme implementation (eight months), different meetings were held with teachers (three two-hour meetings each). However, the teachers kept in contact and the main researcher had continuous contact with them, meeting with them weekly (every Friday) through a phone group to share possible concerns and solutions. Finally, different sessions were recorded and assessed by the research team during the intervention (a minimum of one session was recorded per month, and in one week, they provided a report with possible changes and suggestions).

#### 2.3.2. TPSR Intervention Programme

Hellison’s programme fully realised that getting students to become positive contributors to their community meant that experiences that engender a greater sense of being a responsible person had to be provided [25]. In this regard, Hellison proposed a structured session in five parts. In this study, we joined parts four and five: (1) Relational time: a period where the teacher tries to create a good climate for students. (2) Awareness talk: in this part, the teacher presents the values and the academic content of the lesson. (3) Activity plan: the main part of the lesson; in this sense, this is where the teacher tries to use the different strategies integrated in the tasks. (4) Group meeting and self-reflection time: near the end of the lesson, the students are predisposed to express their opinions about what happened. Here, they use the “thumb technique”, pointing their thumbs up (positive appraisal), to one side (medium) or down (negative appraisal).

#### 2.3.3. Fidelity of Implementation

Hastie et al. [26] suggested that is important to take into account three aspects: (a) a description of the teaching unit, with the different elements, (b) a detailed validation of the model’s implementation based on the pedagogical model, and (c) a detailed description of the programme context. Here, we describe the validation of the programme implementation.

To validate the model’s implementation, one session per teacher and month (two sessions in the first month) was recorded and analysed. The Tool for Assessing Responsibility-based Education (TARE) was used to assess strategies applied to teach responsibility [27]. Four observers analysed the presence or absence of the strategies applied by teachers during the lessons in five-minute periods. The teachers were filmed, and they received a behaviour report and suggestions for improvement. In the control and EG-GE groups, two teachers were selected to represent these groups in the results. They were selected at the beginning of the study, to make at least one record per month. The camera was installed in the classroom prior to the beginning of the study to familiarise students and avoid spontaneous behaviour. By the end of the programme, all teachers used all strategies. Inter-reliability was implemented between the new experts, and a researcher experimented in this kind of analysis. Moreover, the intra-observer was carried out by analysing two different moments over two days, ensuring agreement of over 90%. The checklist instrument was composed by the Tool for Assessing Responsibility-based Education [28]. The instrument was used to identify the responsibility elements respectively. Total agreements (TA) were calculated using the formula: number of total agreements (NTA) divided by agreements (*A*) plus disagreements (*D*) (TA = NTA/*A* + *D*).

### 2.4. Measurements

A closed-question questionnaire was used in the present study. It was divided into two parts—the first one concerned sociodemographic variables, and the second one included the different questionnaires used in the study:(1)Personal and Social Responsibility Questionnaire (PSRQ): to check responsibility values. It was adapted for the Spanish school context by Escartí et al. [29]. It consists of a 14-item scale (seven for personal and seven for social responsibility) with a Likert-type scale from 1 (totally disagree) to 6 (totally agree). The values of reliability were 0.89 for social responsibility and 0.86 for personal responsibility, while they were both 0.085 in the post-test. Reliability of the complete scale was 0.91 in the pre-test and 0.90 in the post-test.(2)Psychological Need Satisfaction in Exercise (PNSE): to check the satisfaction of basic psychological needs (autonomy, competence and relationships). This scale was adapted to the educational and Spanish context by Moreno et al. [30]. This scale consists of 18 items, six to assess each need. These were preceded by the sentence “During my class…” with a Likert-type scale from 1 (false) to 6 (true). The values of reliability were: in the pre-test, 0.72 for autonomy, 0.73 for competence, and 0.75 for relatedness; and in the post-test 0.76, 0.80, and 0.77, respectively. However, an index called the Psychological Mediator Index (PMI) was applied to assess the three variables jointly, with an internal consistency of 0.85 and 0.86 in the pre- and post-test.(3)Motivation towards Education Scale (in French, EME): to check motivation. The Spanish version of the “Échelle de Motivation en Éducation”, validated by Núñez et al. [31], was used. The questionnaire consists of seven subscales, namely, intrinsic motivation to know, accomplish, and experience sensations, identify motivation, introjected motivation, external motivation, and amotivation. The instrument is composed of 28 items preceded by the sentence “I go to school because…” with a seven-point Likert-type scale from 1 (totally disagree) to 7 (totally agree), distributed into seven subscales. The values of internal consistency were: intrinsic motivation to know (0.82 pre-test, 0.81 post-test), experience (0.74 pre-test, 0.80 post-test), and accomplish sensations (0.84 pre-test, 0.85 post-test), general intrinsic motivation (0.90 pre-test, 0.92 post-test), identified regulation (0.69 pre-test, 0.71 post-test), external regulation (0.69 pre-test, 0.73 post-test), introjected regulation (0.80 pre-test, 0.82 post-test), and amotivation (0.78 pre-test, 0.75 post-test). Moreover, the Self-Determination Index (SDI) was applied.(4)Teenage Inventory of Social Skills (TISS): to check prosocial and antisocial behaviour. It was created by Inderbitzen et al. [32] and translated into Spanish by Inglés et al. [33]. The questionnaire has two scales: prosocial behaviour (e.g., participation or altruism) and antisocial behaviour (e.g., social anxiety or insolence). It consists of a six-point Likert-type scale from 1 (does not describe me at all) to 6 (totally describes me). In this study, we used the prosocial scale with an internal consistency of 0.88 in the pre-test and 0.87 in the post-test.(5)Questionnaire on School Violence (in Spanish, CUVE): to check the perception of violence. It was created by Álvarez et al. [34]. It has 44 items and a Likert-type scale from 1 (totally disagree) to 5 (totally agree). The questionnaire’s total internal consistency was 0.97 and 0.96, respectively, in the pre- and post-test.(6)Questionnaire to Assess School Social Climate (CECSCE): to check the climate perceived by students regarding their class, teacher, and school. It was made by Trianes et al. [35]. This questionnaire has two scales: “Climate relative to the school” and “Climate relative to the teachers”, composed of eight and six items, respectively. This scale has a five-point Likert-type scale from 1 (totally disagree) to 5 (totally agree). The Cronbach values (from their internal consistency) were from school climate (0.81 in the pre-test and 0.83 in the post-test) and teacher climate (0.81 in the pre-test and 0.76 in the post-test). The internal consistency for general classroom climate (average of the two climate types) was 0.87 in the pre- and post-test.(7)Resilience Scale (RS-14): designed by Sánchez-Teruel et al. [36] based on the scale created by Wagnild et al. [37] to measure the degree of individual resilience, considered a positive personality characteristic that enhances individual adaptation to adverse situations. The first factor, called personal competence, yielded an internal consistency value of 0.84 in the pre-test and 0.82 in the post-test. The second factor, acceptance of self and life, yielded 0.69 in the pre-test and 0.64 in the post-test. The internal consistency of the complete scale was 0.87 in the pre-test and 0.85 in the post-test.

### 2.5. Ethics

To conduct the study, written consent from the researcher’s University Ethics Committee, the board of directors of the school, and parents/guardians of each student was obtained. The students agreed to participate and were treated in accordance with the ethical guidelines of the American Psychological Association with respect to participant assent, parent/guardian consent, confidentiality, and anonymity. Informed consent was requested from the students and their parents. An introduction letter was addressed to three secondary schools and the approval of the ethics committee was obtained (1685/2017).

### 2.6. Data Analysis

Data analysis was conducted in two consecutive phases. The first phase concerned questionnaire analysis, for which IBM SPSS 22.0 software was used. Questionnaire reliability was analysed by calculating the internal consistency in the pre- and post-test using Cronbach’s alpha test. Before carrying out a MANOVA, Box’s M test of equality of covariance, Bartlett’s and Levene’s test of homogeneity of variance tests were run to check some of the parametric statistical assumptions. Box’s M test data failed (*p* < 0.05), and Bartlett’s (*p* < 0.05) and Levene’s (*p* > 0.05) tests were positive; therefore, not all the starting hypotheses were fulfilled, and it is not possible to consider that the results are conclusive. A repeated-measures MANOVA was carried out to analyse all the dependent variables jointly, where the intra-subject factor was called time (with two levels: pre- and post-test) and the group was considered as the inter-subject factor (with two levels: control and experimental). A residual analysis revealed compliance in most variables of the hypothesis of normality, so it was decided to follow the analyses through these parametric tests. *p*-value and eTa (eta squared) were checked following the recommendations of Richardson [38], considering that small, medium, and large effect sizes were designated as 0.01, 0.06, and 0.14, respectively. The effect size (see Tables 3–9) was calculated to quantify the magnitude of the difference between two means [39]. However, Cohen’s d effect was performed to show the differences for each separate group in the pre-test and post-test. Cohen [39] suggested that d = 0.2 represents a “small” effect size, 0.5 a “medium” effect size, and 0.8 a “large” effect size. Finally, the differences between gender were analysed using a multiple one-way ANOVA.

## 3. Results

### 3.1. Fidelity Results

In Table 2, we can see the fidelity implementation results. In this sense, the EG-GE and EG-PE groups had a percentage of strategy use above 60% for all categories (mean of 64.1% and 67.4%), except for the EG-GE group in “involving students in assessment” and “transfer” and for the EG-PE group in “leadership”, “involving students in assessment”, and “transfer”. This was probably due to these strategies being more appropriate for the final sessions because they require a high level of responsibility (involving students’ assessment and leadership) and the use of transfer happens only at the end of each session. The control group only had a percentage above 60 with respect to modelling, setting expectations, and opportunities for success.

### 3.2. Students Questionnaires

According to the *p*-value, Table 3, Table 4, Table 5, Table 6, Table 7, Table 8 and Table 9 show—in the pre-test analysis—differences in autonomy between the CG and EG-PE and between the EG-GE and EG-PE, in personal responsibility between the CG and EG, and in violence between the CG and EG-PE and between the EG-GE and EG-PE. The EG-GE presented the lowest autonomy and highest violence levels, while the CG showed the lowest personal responsibility levels. Regarding the effect size of univariate analysis from MANOVA, we show a small–moderate effect size in intrinsic motivation (IM) to know, IM to experience, general IM, resilience–competence, total resilience, autonomy, prosocial behaviour, perceived violence, and PMI. But the effect was only near-high for autonomy (0.048) and perceived violence (0.045).

According to the *p*-value, a post-test analysis among the groups was performed (Table 3, Table 4, Table 5, Table 6, Table 7, Table 8 and Table 9) and differences were found between the CG and EG-GE in resilience and its three dimensions, competence, and social responsibility. Significant differences between the CG and EG-PE were only observed in intrinsic motivation (IM) to know. In any case, the results were in favour of the EG-GE or EG-PE, both reaching higher values than the CG. Perceived violence values were lower in the EG-PE than in the other two groups (CG and EG-GE). Regarding the effect size of univariate analysis from MANOVA, we observed a small–moderate effect size in all variables except external R and amotivation. However, a near-moderate effect was shown in IM to experience (−0.056), general IM (0.048), introjected R. (−0.055), teacher climate (−0.055), personal responsibility −0.050), and perceived violence (0.043), and a moderate–high effect was found in autonomy (0.105), school climate (−0.084), PMI (−0.082), and general climate (−0.082).

For the pre- to post-test differences per group (Table 3, Table 4, Table 5, Table 6, Table 7, Table 8 and Table 9) regarding the *p*-value yielded in the CG, there were only changes in PMI. In the EG-GE, differences were found in IM to know, IM to experience, general IM, introjected R., resilience–acceptance, resilience–competence, total resilience, autonomy, competence, teacher climate, prosocial behaviour, social and personal responsibility, SDI, PMI, and general climate. Finally, in the EG-PE, differences were found in introjected R., external R., amotivation, teacher climate, school climate, prosocial behaviour, SDI, and general climate. Cohen’s size [39] shows a small–moderate effect for the EG-GE group in all variables (between 0.2 and 0.5), except for resilience–acceptance (−0.57), total resilience (−0.54), teacher climate (−0.54), and social responsibility (−0.52), which saw a moderate–high effect. The EG-PE group had a small value in most variables, except in IM to accomplish (−0.23), introjected R. (0.23), external R. (−0.27), amotivation (0.28), teacher climate (0.34), school climate (−0.32), prosocial behaviour (−0.28), SDI (−0.28), and general climate (−0.35), which saw a small–moderate effect. Finally, the CG had a small effect in all variables except introjected R. (0.22), resilience–acceptance (−0.23), autonomy (0.46), social relationships (0.28), teacher climate (0.25), school climate (0.29), PMI (0.34), and general climate (0.31).

Finally, pre-test-to-post-test differences between groups (repeated measures, using group as dependent variable) were analysed (Table 3, Table 4, Table 5, Table 6, Table 7, Table 8 and Table 9). The differences yielded *p* < 0.05 for introjected R., competence, resilience-competence, total resilience, social responsibility, and personal responsibility, and *p* < 0.001 for autonomy, PMI, teacher climate, school climate, prosocial behaviours, and general climate. Regarding the effect size, the following variables presented *p* < 0.05: introjected R. (0.028), resilience–competence (0.025), total resilience (0.028), autonomy (0.073), competence (0.023), teacher climate (0.061), school climate (0.045), prosocial behaviour (0.032), social responsibility (0.023), personal responsibility (0.028), PMI (0.055), and general climate (0.062).

Differences between gender were checked using a multiple one-way ANOVA on the pre-test and post-test variables. Among female students, there were only differences in the pre-test in competence between the EG-PE and EG-GE (in favour of the EG-PE; *p* = 0.032). There were also differences in the post-test between the CG and EG-GE in favour of the EG-GE in IM to know (*p* = 0.004), IM to experience (*p* = 0.003), IM to accomplish (*p* = 0.017), general IM (*p* = 0.002), introjected R. (*p* = 0.001), resilience–acceptance (*p* = 0.030), autonomy (*p* < 0.001), teacher climate (*p* = 0.021), school climate (*p* = 0.035), personal responsibility (*p* = 0.040), IMP (*p* = 0.002), and general climate (*p* = 0.012). The EG-PE group showed differences in the CG, in favour of the EG-PE in IM to experience (*p* = 0.025), autonomy (*p* = 0.002), teacher climate (*p* = 0.002), school climate (*p* < 0.001), and general climate (*p* < 0.001).

Among male students, there were differences in the pre-test between the CG and EG-PE in favour of the EG-PE in resilience–competence (*p* = 0.026), total resilience (*p* = 0.026), and personal responsibility (*p* = 0.001). There was one difference between the EG-PE and EG-GE in favour of the EG-PE in autonomy (*p* = 0.041). Finally, there were differences among the EG-GE, CG (*p* < 0.001), and EG-PE (*p* < 0.001), always in favour of the EG-GE in violence. In the post-test, there were differences between the EG-PE and CG in favour of the EG-PE in introjected regulation (*p* = 0.037), autonomy (*p* = 0.039), and personal responsibility (*p* = 0.039). However, the EG-GE group showed differences with the CG, in favour of the EG-GE in the post-test in autonomy (*p* = 0.016) and personal responsibility (*p* = 0.012).

## 4. Discussion

The main aim of the present study was to examine the effects of the TPSR model’s implementation in a secondary school’s general context (EG-GE) on responsibility, motivation, satisfaction of basic psychological needs, resilience, prosocial behaviour, violence, and classroom climate, compared with an EG-PE (which is how the model has been traditionally applied) and a CG, and to seek the differences according to gender.

From the questionnaire analysis, it can be highlighted that the EG-GE experienced a statistically significant improvement in personal and social responsibility, suggesting that the model’s implementation duration was appropriate [37]. This is also in agreement with the positive results in responsibility when the duration of the study using TPSR is sufficient [40].

Considerable improvements were found in the self-determined motivation for both the EG-GE and EG-PE, as well as in satisfaction of basic psychological needs for the EG-GE. These results were similar to those obtained by Manzano-Sánchez et al. [17] in the most recent studies involving these variables in all educational contexts, especially with primary and secondary students. Moreover, Paväo et al. [41] applied this methodology in preschool students as the first study to do as such, and they concluded that TPSR could be adapted and implemented in this context.

The TPSR model’s implementation also led to highly positive results in prosocial behaviour, unlike the study by García-García et al. [42], who found no improvement with the application of TPSR to this variable. However, Pascual et al. [23] showed a benefit towards teachers’ perception of the behaviour of their students. Furthermore, general climate, especially classroom climate, improved substantially in the EG-GE and EG-PE. This was in keeping with other studies that include this variable, such as the study by Manzano-Sánchez et al. [12] or in an observational analysis by Camerino et al. [24]. The latter was the first to apply TPSR to all school subjects and found increases in self-determined motivation, autonomy, and classroom climate, results that are markedly similar to those of the present study.

By contrast, violence decreased in the EG-GE, but not to a statistically significant extent. More emphasis on this aspect would be needed in order to obtain an improvement in this variable. These authors found significant improvements in attitudes towards violence, social responsibility, competence, and relationships among participants. However, other authors [42] found no significant improvement either in suffered or perceived violence.

Lastly, resilience is one of the major novelties of this study, since no previous study had measured it after the application of TPSR or other educational programmes. We concluded that TPSR may be effective at improving students’ resilience. However, it would be essential to apply it to all school subjects, since no improvements were found in the group where it was applied in PE only. In this field, the study by Ungar et al. [43] concluded that it would be necessary to include school and family contexts in the different interventions in order to improve resilience in children.

According to gender, focussing on girls, both experimental groups saw improved IM to experience, autonomy, and all climate-related variables. However, in the EG-GE, general IM, IM to know, IM to accomplish, introjected regulation, resilience–acceptance, personal responsibility, and IMP were also improved. As for the boys, it should be noted that the EG-PE started out with higher levels of autonomy than the CG, while in the post-test, there were significant differences for both the EG-PE and EG-GE versus the GC. These differences are in accordance with the study by Manzano-Sánchez et al. [17], who showed the benefits of the TPSR programme only in female students. However, this study saw differences in personal responsibility after the programme in male and female students—as was the case in the present study—but not in social responsibility, probably because most values of social responsibility are in the last level of the programme and it was applied at the end of the intervention. It is interesting to highlight that Gómez-Mármol et al. [44] found that males had higher values of responsibility, and this was related with greater practice of physical activity. Showing the rest of the variables, female students saw improved values for teacher and student climate in both groups, but only the EG-GE group improved their resilience. Among male students, there were no differences in these variables, except in violence in favour of the EG-GE. This seems to go against the results of Sánchez-Alcaraz et al. [15], who saw that male students experience improved psychological measurements more so than girls. However, the study by Carbonero et al. [45]—a two-year longitudinal study—concluded that male students had higher values in different aspects such as friendliness, tidiness, and being well-mannered. However, both boys and girls had a higher level of academic achievement.

Therefore, taking into account the results given in the detailed analysis leads us to consider the importance of making students aware of their behaviour in order to achieve measurable changes in their conduct (with significant improvements in the different psychosocial variables produced over time in the experimental groups). Along the same lines, Manzano-Sánchez et al. [17] corroborated the importance of coordination and involvement of all teachers participating in the application of TPSR. In addition, Martinek et al. [46] highlighted that the future of TPSR must take into account the contributions of teachers and instructors in order to achieve adequate development and their commitment to giving the best of themselves [40]. However, future studies can continue expanding the Hellison Model in an international context, since nowadays, a total of 31 countries have reported some level of the presence of TPSR [47], including recent studies with preservice teachers who implemented this methodology and evaluated it positively [48].

As regards limitations, it is important to underline that the selection was based on convenience. It is relevant for future research to try to check the effects of the programme at a midpoint (with an intra-test). However, a longitudinal study (in two full-year courses) may be interesting to check the results over time. Prospective studies should consider involving families as the major determinant of student responsibility [49].

## 5. Conclusions

Using TPSR for at least 60% of the total teaching time with secondary school students led to overall improvements in responsibility, the self-determined motivation, satisfaction of basic psychological needs, resilience, prosocial behaviour, and classroom climate, as well as a positive trend in perceived violence.

TPSR results may be optimised through application to the whole educational context, modifying the traditional PE-only implementation pattern, which produced positive results, but not in all variables under study. Considering gender, female students may have better results than male students, in both cases showing better results in a group with more subjects implementing TPSR than only in PE.

Future lines of research could combine a mixed-method programme and include new variables to study. It could also be interesting to check if the results are similar or different between primary and secondary school. Moreover, continuing with Vallerand’s sequence, showing the casual relationships between responsibility, basic psychological needs, motivation, school climate, prosocial behaviour, resilience, or violence could be valuable. However, several studies have combined this model with others (e.g., Cooperative Learning or Game-Centred Approach) [50,51,52], which could help obtain better results.

## Figures and Tables

**Table 1 ijerph-18-03047-t001:** Task examples in three different subjects.

	Physical Education	Spanish Language	Mathematics
**Level 1**	Play a basketball game where everyone has to touch the ball before throwing it.	Syntax learning in groups of four. Together, form five sentences, with obligatory participation, without interrupting their classmates.	In groups of six, solve simple mathematical operations; not all participants are able to participate, and they are instructed to always encourage their classmates.
**Level 2**	Perform a strength and endurance circuit using the Borg scale according to the level they are at (1 to 10).	Do morphology exercises with three levels of difficulty, with each student choosing the one that is a challenge for them.	Perform a series of exercises of second-degree problem equations where each person should try to do all the exercises they can, where all students receive a point for trying; these points are then added together in a collective mark for the whole class.
**Level 3**	Make a personal plan to work on stretching a muscle group after carrying out an assessment.	Prepare an essay on any author from the “Generation of ’98” and present it in class.	Based on what they have perceived as more difficult, do a project with a proposal to improve the content and example exercises (rule of three, second-degree equations, basic geometric problems, etc.).
**Level 4**	Peer teaching: one student teaches another how to do a hand touch in volleyball, correcting them after they have learnt it.	In groups of four, each person writes part of a text commentary, where one is the leader who decides who prepares and presents what part to the rest of the class.	Everyone is grouped in pairs. One student performs a math problem (calculation of areas) at home, and then in the next lesson, they present it to their partner and help them solve it; their partner does the same with another topic (calculation of the hypotenuse).
**Level 5**	Acting as a referee in a handball championship with a low grade.	With the youngest students, build a giant goose game with language and literature questions where, in pairs, the older students can help the younger ones, but only with gestures.	In groups of six, make a popular market with made-up products where parents can go shopping and the students are the guides and workers.

**Table 2 ijerph-18-03047-t002:** Fidelity Implementation Instrument.

	EG-GE	EG-PE	Control
	Class Use	Class Use	Class Use
**“Modelling respect” (M). Teacher models respectful communication. This would involve communication with the whole group and individual students.**	1.00	1.00	0.97
**“Setting expectations” (E). Teacher explains or refers to explicit behavioural expectations. These could relate to safe practices, rules and procedures, or etiquette.**	0.78	0.9	0.76
**“Opportunities for success” (S). Teacher structures lessons so that all students have the opportunity to successfully participate and be included regardless of individual differences.**	0.79	0.7	0.87
**“Fostering social interaction” (SI). Teacher structures activities to foster positive social interaction. This could involve student–student interaction through cooperation, teamwork, problem solving, conflict resolution, or debriefing.**	0.65	0.92	0.04
**“Assigning tasks” (T). Teacher assigns specific responsibilities or tasks that facilitate the organisation of the programme or a specific activity. This could manifest as taking attendance, setting up equipment, keeping scores/records, or officiating a game.**	0.67	0.81	0.02
**“Leadership” (L). Teacher allows students to lead or be in charge of a group. This could manifest as demonstrating for the class, leading a station, teaching/leading exercises for the whole class, or coaching a team.**	0.68	0.41	0.26
**“Giving choices and voices” (V). Teacher gives students a voice in the programme. This could involve group discussions, voting as a group, individual choices, students asking questions, making suggestions, sharing opinions, appraising the teacher or programme.**	0.62	0.81	0.31
**“Involving students in assessment” (A). Teacher allows student to have a role in learner assessment. This could take the form of self- or peer-assessment related to skill development, behaviour, attitude, etc.**	0.38	0.31	0.01
**“Transfer” (Tr). Teacher directly addresses the transfer of like skills or responsibilities from the lesson beyond the programmer.**	0.20	0.21	0.01
**Total**	64.1%	67.4%	37.1%

EG-GE = Experimental group in general education; EG-PE = Experimental group in Physical Education.

**Table 3 ijerph-18-03047-t003:** Self-determination theory results (motivation).

s	Group	Pre-Test		Post-Test		Pre–Post Difference		Between-Group Difference of Means		
		Mean	SD	Mean	SD	*p*-Value	Cohen’s d	diff	*p*-Value	eTa
IM to Know	Control	4.82	1.22	4.68	1.33	0.475	0.11	0.138	0.086	0.019
	Experimental	4.78	1.02	5.16	1.28	0.020 *	−0.33	−0.382		
	PE Experimental	5.16	1.30	5.24	1.18	0.517	−0.07	−0.086		
	*p*-value + eTa	0.059	0.022	0.014 *	0.033					
IM to Experience	Control	3.95	1.34	3.74	1.44	0.340	0.15	0.215	0.065	0.021
	Experimental	4.02	1.13	4.42	1.50	0.021 *	−0.30	−0.405		
	PE Experimental	4.35	1.32	4.58	1.32	0.142	−0.18	−0.231		
	*p*-value + eTa	0.079	0.020	0.001 **	0.056					
IM to Accomplish	Control	4.75	1.26	4.76	1.42	0.962	−0.01	−0.010	0.441	0.006
	Experimental	4.95	1.03	5.30	1.38	0.060	−0.28	−0.345		
	PE Experimental	4.98	1.38	5.27	1.14	0.057	−0.23	−0.289		
	*p*-value + eTa	0.470	0.006	0.021 *	0.030					
General IM	Control	4.51	1.16	4.40	1.25	0.555	0.09	0.110	0.093	0.019
	Experimental	4.58	0.91	4.96	1.29	0.017 *	−0.34	−0.377		
	PE Experimental	4.83	1.15	5.03	1.07	0.090	−0.18	−0.202		
	*p*-value + eTa	0.119	0.017	0.002 **	0.048					
Identified R.	Control	5.34	1.01	5.26	1.27	0.692	0.07	0.076	0.304	0.009
	Experimental	5.40	0.93	5.67	1.22	0.088	−0.25	−0.271		
	PE Experimental	5.40	1.06	5.52	0.94	0.294	−0.12	−0.121		
	*p*-value + eTa	0.917	0.001	0.088	0.019					
Introjected R.	Control	4.97	1.37	4.65	1.57	0.229	0.22	0.321	0.028 *	0.028
	Experimental	5.06	0.97	5.45	1.48	0.043 *	−0.32	−0.400		
	PE Experimental	5.07	1.44	5.37	1.15	0.028 *	−0.23	−0.303		
	*p*-value + eTa	0.889	0.001	0.001 **	0.055					
External R.	Control	5.73	1.02	5.82	1.05	0.637	−0.09	−0.090	0.652	0.003
	Experimental	5.79	1.03	6.07	1.01	0.056	−0.28	−0.280		
	PE Experimental	5.73	0.94	5.98	0.90	0.033 *	−0.27	−0.246		
	*p*-value + eTa	0.910	0.001	0.304	0.009					
Amotivation	Control	2.17	1.34	2.02	1.24	0.480	0.11	0.142	0.764	0.002
	Experimental	2.39	1.01	2.11	1.47	0.155	0.22	−0.280		
	PE Experimental	2.28	1.45	1.94	0.97	0.042 *	0.28	0.341		
	*p*-value + eTa	0.566	0.004	0.658	0.003					
SDI	Control	4.68	3.81	4.82	4.46	0.829	−0.03	0.140	0.310	0.009
	Experimental	4.37	2.91	5.76	4.85	0.022 **	−0.35	−1.389		
	PE Experimental	5.10	4.07	6.16	3.39	0.023 **	−0.28	−1.055		
	*p*-value + eTa	0.377	0.009	0.131	0.016					

Note: * *p* < 0.05; ** *p* < 0.01; eTa = effect size of MANOVA; Cohen’s d = Cohen effect size. This effect size was marked as small (0.2), medium (0.5), and large (0.8) [39]. IM = Intrinsic Motivation; R = Regulation; SDI = Self-Determination Index.

**Table 4 ijerph-18-03047-t004:** Self-determination theory results (psychological basic needs).

	Group	Pre-Test		Post-Test		Pre–Post Difference		Between-Group Difference of Means		
		Mean	SD	Mean	SD	*p*-Value	Cohen’s d	diff	*p*-Value	eTa
Autonomy	Control	3.39	0.65	3.06	0.77	0.009 **	0.46	0.325	<0.001 **	0.073
	Experimental	3.35	0.69	3.70	0.81	0.002 **	−0.47	−0.350		
	PE Experimental	3.70	0.80	3.59	0.73	0.256	0.13	0.103		
	*p*-value + eTa	0.002 **	0.048	<0.001 **	0.105					
Competence	Control	3.64	0.67	3.57	0.81	0.581	0.09	0.070	0.050 *	0.023
	Experimental	3.64	0.68	3.94	0.92	0.021 *	−0.37	−0.300		
	PE Experimental	3.82	0.71	3.80	0.73	0.835	0.03	0.020		
	*p*-value + eTa	0.135	0.016	0.023 *	0.029					
Social Relationships	Control	4.00	0.76	3.78	0.86	0.142	0.28	0.225	0.087	0.019
	Experimental	3.93	0.58	4.07	0.82	0.170	−0.20	−0.142		
	PE Experimental	3.95	0.84	3.94	0.76	0.960	0.01	0.005		
	*p*-value + eTa	0.826	0.002	0.080	0.020					
PMI	Control	3.68	0.58	3.47	0.65	0.045 *	0.34	0.207	0.001 **	0.055
	Experimental	3.64	0.52	3.90	0.66	0.004 **	−0.44	−0.264		
	PE Experimental	3.82	0.66	3.78	0.61	0.549	0.07	0.042		
	*p*-value + eTa	0.089	0.019	<0.001 **	0.082					

Note: * *p* < 0.05; ** *p* < 0.01; eTa = effect size of MANOVA; Cohen’s d = Cohen effect size. This effect size was marked as small (0.2), medium (0.5), and large (0.8) [39]. PMI = Psychological Mediator Index.

**Table 5 ijerph-18-03047-t005:** Resilience results.

	Group	Pre-Test		Post-Test		Pre–Post Difference		Between-Group Difference of Means		
		Mean	SD	Mean	SD	*p*-Value	Cohen’s d	diff	*p*-Value	eTa
Resilience–Acceptance	Control	4.68	1.19	4.94	1.06	0.147	−0.23	−0.262	0.058	0.022
	Experimental	4.83	0.93	5.42	1.14	0.001 **	−0.57	−0.588		
	PE Experimental	4.94	1.17	5.04	1.03	0.473	−0.09	−0.099		
	*p*-value + eTa	0.313	0.009	0.012 *	0.034					
Resilience–Competence	Control	5.00	0.93	5.10	1.03	0.487	−0.11	−0.108	0.041 *	0.025
	Experimental	5.10	0.77	5.50	0.90	0.002 **	−0.48	0.399		
	PE Experimental	5.32	0.89	5.29	0.83	0.811	0.03	0.026		
	*p*-value + eTa	0.050	0.023	0.025 *	0.029					
Total Resilience	Control	4.93	0.93	5.07	0.94	0.343	−0.15	−0.141	0.028 *	0.028
	Experimental	5.04	0.74	5.48	0.89	<0.001 **	−0.54	−0.439		
	PE Experimental	5.24	0.89	5.24	0.81	0.992	0.00	−0.001		
	*p*-value + eTa	0.063	0.022	0.012 *	0.034					

Note: * *p* < 0.05; ** *p* < 0.01; eTa = effect size of MANOVA; Cohen’s d = Cohen effect size. This effect size was marked as small (0.2), medium (0.5), and large (0.8) [39].

**Table 6 ijerph-18-03047-t006:** Social school climate results.

	Group	Pre-Test		Post-Test		Pre–Post Difference		Between-Group Difference of Means		
		Mean	SD	Mean	SD	*p*-Value	Cohen’s d	diff	*p*-Value	eTa
Teacher Climate	Control	3.62	0.75	3.43	0.83	0.141	0.25	0.197	<0.001 **	0.061
	Experimental	3.49	0.61	3.83	0.66	<0.001 **	−0.54	−0.342		
	PE Experimental	3.58	0.79	3.85	0.79	<0.001 **	−0.34	−0.271		
	*p*-value + eTa	0.458	0.006	0.001 **	0.055					
School Climate	Control	3.33	0.76	3.12	0.69	0.088	0.29	0.214	0.003 *	0.045
	Experimental	3.30	0.58	3.45	0.83	0.126	−0.20	−0.145		
	PE Experimental	3.46	0.73	3.69	0.73	0.001 **	−0.32	−0.232		
	*p*-value + eTa	0.259	0.006	<0.001 **	0.084					
General Climate	Control	3.46	0.68	3.25	0.66	0.077	0.31	0.207	<0.001 **	0.062
	Experimental	3.38	0.55	3.61	0.67	0.006 **	−0.38	−0.230		
	PE Experimental	3.51	0.70	3.76	0.72	<0.001 **	−0.35	−0.248		
	*p*-value + eTa	0.391	0.007	<0.001 **	0.082					

Note: * *p* < 0.05; ** *p* < 0.01; eTa = effect size of MANOVA; Cohen’s d = Cohen effect size. This effect size was marked as small (0.2), medium (0.5), and large (0.8) [39].

**Table 7 ijerph-18-03047-t007:** Prosocial behaviour results.

	Group	Pre-Test		Post-Test		Pre–Post Difference		Between-Group Difference of Means		
		Mean	SD	Mean	SD	*p*-Value	Cohen’s d	diff	*p*-Value	eTa
Prosocial Behaviour	Control	4.46	0.71	4.33	0.70	0.318	0.17	0.122	0.016 *	0.032
	Experimental	4.32	0.55	4.58	0.79	0.003 **	−0.38	−0.260		
	PE Experimental	4.29	0.83	4.51	0.73	0.014 *	−0.28	−0.219		
	*p*-value + eTa	0.314	0.011	0.117	0.017					

Note: * *p* < 0.05; ** *p* < 0.01; eTa = effect size of MANOVA; Cohen’s d = Cohen effect size. This effect size was marked as small (0.2), medium (0.5), and large (0.8) [39].

**Table 8 ijerph-18-03047-t008:** Personal and social responsibility results.

	Group	Pre-Test		Post-Test		Pre–Post Difference		Between-Group Difference of Means		
		Mean	SD	Mean	SD	*p*-Value	Cohen’s d	diff	*p*-Value	eTa
Social Responsibility	Control	4.75	0.85	4.81	0.81	0.684	−0.07	−0.054	0.050 *	0.023
	Experimental	4.74	0.70	5.13	0.82	<0.001 **	−0.52	−0.391		
	PE Experimental	4.99	0.89	5.05	0.80	0.546	−0.07	−0.062		
	*p*-value + eTa	0.076	0.009	0.039 *	0.025					
Personal Responsibility	Control	4.57	0.96	4.60	1.04	0.878	−0.03	−0.025	0.026 *	0.028
	Experimental	4.74	0.72	5.12	0.88	0.002 **	−0.47	−0.375		
	PE Experimental	5.03	0.92	4.98	0.82	0.622	0.06	0.048		
	*p*-value + eTa	0.003 **	0.020	0.001 **	0.050					

Note: * *p* < 0.05; ** *p* < 0.01; eTa = effect size of MANOVA; Cohen’s d = Cohen effect size. This effect size was marked as small (0.2), medium (0.5), and large (0.8) [39].

**Table 9 ijerph-18-03047-t009:** Resilience, social school climate, prosocial behaviour, and perceived violence results.

	Group	Pre-Test		Post-Test		Pre–Post Difference		Between-Group Difference of Means		
		Mean	SD	Mean	SD	*p*-Value	Cohen’s d	diff	*p*-Value	eTa
Perceived Violence	Control	2.19	0.85	2.33	0.92	0.365	−0.16	−0.139	0.185	0.013
	Experimental	2.44	0.61	2.32	0.72	0.237	0.18	0.128		
	PE Experimental	1.88	0.78	2.00	0.68	0.208	−0.16	−0.115		
	*p*-value + eTa	<0.001 **	0.045	0.004 **	0.043					

Note: ** *p* < 0.01; eTa = effect size of MANOVA; Cohen’s d = Cohen effect size. This effect size was marked as small (0.2), medium (0.5), and large (0.8) [39].

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
