# Peer review of "Application of the Teaching Personal and Social Responsibility Model in the Secondary Education Curriculum: Implications in Psychological and Contextual Variables in Students"

_ijerph, 2021, doi:10.3390/ijerph18063047_

Round 1
Reviewer 1 Report
The authors aim to apply teaching personal and social responsibility (TPSR) model to the Secondary Education curriculum and examine the effects on responsibility, motivation, satisfaction of basic psychological needs, resilience, prosocial behaviours, violence and classroom climate.
As the key of the study was the implementation of TPSR, the arthors are advised to add the rationale of the design and implementation of the TPSR programme, e.g. why 60% ot the total teaching time, why the activities adopted for the programme and control (e.g. Table 1)...
The manuscript has to be reviewed and improved significantly to improve its English readability, clarity and organisation.
The aim of the study as shown in the abstract was not clearly illustrated and confusing. "The aim was to implement a value-promoting program (Teaching Personal and Social Responsibility, TPSR) and to assess its impact on students.". The same applies to the aims shwon in the introduction (line 77-78) and the discussion (line 310).
As the title of manuscript implies, what were the implications for students and teachers? Illustrate them clearly in the discussion and conclusion.
The discussion was too brief and has to be enhanced and enriched and it requires substantiation of relevant literature and supporting evidence.
Author Response
The authors aim to apply teaching personal and social responsibility (TPSR) model to the Secondary Education curriculum and examine the effects on responsibility, motivation, satisfaction of basic psychological needs, resilience, prosocial behaviours, violence and classroom climate.
As the key of the study was the implementation of TPSR, the arthors are advised to add the rationale of the design and implementation of the TPSR programme, e.g. why 60% ot the total teaching time, why the activities adopted for the programme and control (e.g. Table 1).
- It has added the rationale of the design and implementation of the TPSR programme, explaining why it was the 60% of the total teaching time.
- The activities adopted for the programme come from Manzano-Sánchez, Merino-Barrero, Sánchez-Alcaraz and Valero-Valenzuela (2020) book. It has added this reference in the Design and References sections.
The manuscript has to be reviewed and improved significantly to improve its English readability, clarity and organisation.
- The services of an expert in translation has been required and the manuscript has been proofreading.
The aim of the study as shown in the abstract was not clearly illustrated and confusing. "The aim was to implement a value-promoting program (Teaching Personal and Social Responsibility, TPSR) and to assess its impact on students.". The same applies to the aims shwon in the introduction (line 77-78) and the discussion (line 310).
- It has been clarified in abstract, introduction and discussion sections.
As the title of manuscript implies, what were the implications for students and teachers? Illustrate them clearly in the discussion and conclusion.
- It has been removed teachers from the manuscript. It was a mistake, this variable was not measured.
The discussion was too brief and has to be enhanced and enriched and it requires substantiation of relevant literature and supporting evidence.
- Discussion section has been enriched with new studies and references.
Reviewer 2 Report
The study submitted here seeks to analyze the effects of a training program on personal and social responsibility in a group of students (n = 257) of Secondary Education, distributed in a control group and two experimental groups (one receiving at least 60% of the program, and the other -physical education- receiving 100%). To this end, seven validated instruments are applied, covering the different dimensions of the theoretical construct underlying the object of research. In addition, the values of their internal consistency at each point of application (both pretest and posttest) are provided in a timely manner. Overall, the study could contribute to increase the body of knowledge on the development of skills in personal and social responsibility, an already consolidated line of research.
However, it is necessary for the authors to review the following issues in order to improve the quality and scientific soundness of their manuscript:
1. Section 2.6. This section should be specified more precisely. Please, justify the reasons why two types of ANOVA (one-way ANOVA and MANOVA) are used. We remind the authors that a MANOVA is only an ANOVA with two or more continuous response variables. The application of a univariate ANOVA would only make sense if the outcome of the program has not had a positive effect on any of the dependent variables (scales and dimensions of the instruments administered), in which case it would be advisable to analyze whether the training program has affected any of the variables separately. If what is desired, and this is what seems to have been done, is to compare the means of all the dependent variables jointly (MANOVA) and, at the same time, to analyze the influence of the fixed factor (group) on each indicator of all the dimensions that make up the scales applied, this circumstance should be specified, with the required detail, in this subsection.
2. Effect size in MANOVA (ƞ2p). In the results table these values are included (in this study, eTa?), however, they are not commented or analyzed. Please, interpret the magnitude of the values obtained accordingly. Also, clarify the meaning and the particular arrangement in Table 3 of the p-value + eTa (?). This should be clarified either in section 2.6. or, preferably, in the table.
3. Please, incorporate empirical evidence that demonstrates the fulfillment of the parametric statistical assumptions (assumptions for ANOVA and MANOVA): a) Data from simple random samples; b) Existence of normality in their distribution; c) Existence of equal subpopulation variances.
4. Considering the abundant scientific literature available on the concept under study from multiple fields of knowledge (training programs on personal and social responsibility), it is strongly recommended to deepen, expand and update the scientific literature reviewed.
5. Finally, a detailed review of the English grammar by a native speaker is recommended.
Author Response
The study submitted here seeks to analyze the effects of a training program on personal and social responsibility in a group of students (n = 257) of Secondary Education, distributed in a control group and two experimental groups (one receiving at least 60% of the program, and the other -physical education- receiving 100%). To this end, seven validated instruments are applied, covering the different dimensions of the theoretical construct underlying the object of research. In addition, the values of their internal consistency at each point of application (both pretest and posttest) are provided in a timely manner. Overall, the study could contribute to increase the body of knowledge on the development of skills in personal and social responsibility, an already consolidated line of research.
However, it is necessary for the authors to review the following issues in order to improve the quality and scientific soundness of their manuscript:
- Section 2.6. This section should be specified more precisely. Please, justify the reasons why two types of ANOVA (one-way ANOVA and MANOVA) are used. We remind the authors that a MANOVA is only an ANOVA with two or more continuous response variables. The application of a univariate ANOVA would only make sense if the outcome of the program has not had a positive effect on any of the dependent variables (scales and dimensions of the instruments administered), in which case it would be advisable to analyze whether the training program has affected any of the variables separately. If what is desired, and this is what seems to have been done, is to compare the means of all the dependent variables jointly (MANOVA) and, at the same time, to analyze the influence of the fixed factor (group) on each indicator of all the dimensions that make up the scales applied, this circumstance should be specified, with the required detail, in this subsection.
- In section 2.6. it has been explained the statistical tests used to analyse the data in a more precise way, detailing every factor and the decision taken.
- Effect size in MANOVA (ƞ2p). In the results table these values are included (in this study, eTa?), however, they are not commented or analyzed. Please, interpret the magnitude of the values obtained accordingly. Also, clarify the meaning and the particular arrangement in Table 3 of the p-value + eTa (?). This should be clarified either in section 2.6. or, preferably, in the table.
- It has been added Richardson and Cohen references to clarify the effect sizes and theirs interpretation. Therefore, in the result section new comments referring the magnitude of the values has been incorporated.
- Please, incorporate empirical evidence that demonstrates the fulfillment of the parametric statistical assumptions (assumptions for ANOVA and MANOVA): a) Data from simple random samples; b) Existence of normality in their distribution; c) Existence of equal subpopulation variances.
- A serie of tests to check the hypothesis has been developed and detailed in section 2.6.
- Considering the abundant scientific literature available on the concept under study from multiple fields of knowledge (training programs on personal and social responsibility), it is strongly recommended to deepen, expand and update the scientific literature reviewed.
- Discussion section has been enriched with new studies and references.
- Finally, a detailed review of the English grammar by a native speaker is recommended.
- The services of an expert in translation has been required and the manuscript has been proofreading.
Reviewer 3 Report
This article evaluates the effects of an eight-month educational intervention based on the teaching model of personal and social responsibility in three secondary schools.
The objectives set out in the article are interesting. They can provide relevant evidence on this educational methodology.
Nevertheless, I think that authors should adress some important changes that I expose below:
1. Statistical analysis.
1.1. How many MANOVAs have been conducted?
According to table 3, it seems that only one MANOVA has been conducted in which all the subscales of all the research instruments have been compared.
Making so many comparisons at once increases the risk of making a type I error.
A different MANOVA should be performed for each instrument.
1.2. It must be verified that the data fit the normal distribution in order to decide whether to perform parametric or non-parametric statistics.
This check must be done independently for each instrument.
2. Presentation of the results.
2.1. Table 3 offers an unmanageable amount of information.
It is recommended to divide it into several tables, one for each instrument used.
2.2. Even if the results of each instrument are analyzed separately in Table 3, it is difficult to understand the information.
The row for p-value + eTa matches the columns for Mean and SD. This greatly complicates the interpretation of the table.
A simpler way to represent the information should be sought.
3. Results analysis.
The effect size values are very low in all cases, which increases the suspicion that a type I error is being incurred.
It is recommended that the authors take these very low values into account when interpreting the results (that is, do not pay attention solely to the p-values).
Once the authors have reformulated Table 3 and taken into account the effect size values, the results of the study and the differences between groups and between the pre and post test can be better interpreted.
4. Study aims.
It is recommended to reformulate the objectives of the study so that they are more easily understood.
As expressed in lines 77-82, they are too generic aims.
5. Participants.
It should be described in more detail:
5.1. The characteristics of the schools (size, location, socioeconomic level of the environment, etc.). This information should only be located in section the Participants section.
5.2. The demographic characteristics of the teachers (age, years of experience, subjects taught, etc.)
5.3. It should be analyzed whether there are statistically significant differences in the age and sex distribution of the three groups.
6. Article structure.
I'm not sure if the information of table 2 is correctly located in the Materials and Methods section.
I suggest authors to consider if it would be better considered as a result.
Anyway, it's suggested to explain the content of lines 258-259 before this table.
7. Formal issues.
7.1. The text must be revised from the linguistic point of view by a native translator.
7.2. When expressing p-values, avoid indicating .000; <.001 should be used.
7.3. Keywords must be in alphabetic order.
7.4. Line 41. The first time the acronym TPSR appears, its meaning must be indicated. It is recommended to include it in line 32 the first time its meaning appears.
Author Response
This article evaluates the effects of an eight-month educational intervention based on the teaching model of personal and social responsibility in three secondary schools.
The objectives set out in the article are interesting. They can provide relevant evidence on this educational methodology.
Nevertheless, I think that authors should adress some important changes that I expose below:
- Statistical analysis.
1.1. How many MANOVAs have been conducted?
According to table 3, it seems that only one MANOVA has been conducted in which all the subscales of all the research instruments have been compared.
Making so many comparisons at once increases the risk of making a type I error.
A different MANOVA should be performed for each instrument.
We are sorry to disagree in this point. Regarding the second statement, we regret to tell you that you are not correct in it, since, by construction, the MANOVA is designed so that the size of type I error is the value of alpha previously fixed by the researcher. Something similar occurs with the ANOVA with respect to comparing the mean equality between the groups two by two. It is in the latter case that the type I error increases. For this reason, the fact of doing a single MANOVA with all the dependent variables simultaneously guarantees the size of the type I error, since the test is unique. On the contrary, if it is divided and several MANOVAs are carried out, each one with a type I error of 5% and the results are added at the end, the Type I error is not controlled.
For example, if you make 5 independent MANOVAS, each of them with a probability of accepting the null hypothesis when it is true of 95%, in the end the probability of not rejecting the hypothesis is: probability of not rejecting it in the first by probability of not rejecting it in the second by ... by probability of not rejecting it in the fifth = 0.95^5 =0.7737 and, in consequence, alpha=1-0.7737=0.2262 (see, for instance, Rencher, A. C. Methods of multivariate analysis, chapter 6. Ed Wiley, 2002)
- 2. It must be verified that the data fit the normal distribution in order to decide whether to perform parametric or non-parametric statistics.
This check must be done independently for each instrument.
A serie of tests to check the hypothesis has been developed and detailed in section 2.6.
- Presentation of the results.
2.1. Table 3 offers an unmanageable amount of information.
It is recommended to divide it into several tables, one for each instrument used.
Thank you for your suggestion. We have splitted the table 3 in two (table 3: Self-determination theory results and table 4: resilience, social school climate, prosocial behaviors and perceived violence results) with the intention to be more clear.
2.2. Even if the results of each instrument are analyzed separately in Table 3, it is difficult to understand the information.
The row for p-value + eTa matches the columns for Mean and SD. This greatly complicates the interpretation of the table.
A simpler way to represent the information should be sought.
- We understand it is complex to interpretate the tables due to the big amount of information, but on the other hand it is also the simplest way to have all of them in only one table. This is the same procedure we have used in previous manuscripts, but we will be delighted if you propose an alternative.
- Results analysis.
The effect size values are very low in all cases, which increases the suspicion that a type I error is being incurred.
It is recommended that the authors take these very low values into account when interpreting the results (that is, do not pay attention solely to the p-values).
- Your consideration has been taking into account and comments have been added in three different paragraphs in results section.
Once the authors have reformulated Table 3 and taken into account the effect size values, the results of the study and the differences between groups and between the pre and post test can be better interpreted.
We expect we have achieved this with the changes incorporated.
- Study aims.
It is recommended to reformulate the objectives of the study so that they are more easily understood.
As expressed in lines 77-82, they are too generic aims.
- It has been clarified in abstract, introduction and discussion sections.
- Participants.
It should be described in more detail:
5.1. The characteristics of the schools (size, location, socioeconomic level of the environment, etc.). This information should only be located in section the Participants section.
- It has added this information in section 2.2. participants.
5.2. The demographic characteristics of the teachers (age, years of experience, subjects taught, etc.)
- It has added this information in section 2.2. participants.
5.3. It should be analyzed whether there are statistically significant differences in the age and sex distribution of the three groups.
- It has calculated and added this information in section 2.2. participants.
- Article structure.
I'm not sure if the information of table 2 is correctly located in the Materials and Methods section.
I suggest authors to consider if it would be better considered as a result.
Anyway, it's suggested to explain the content of lines 258-259 before this table.
- Se puede incorporar así, no lo he cambiado por no liar las líneas. El comentario de explicar los contenidos de las líneas 258-259 no lo entiendo
- Formal issues.
7.1. The text must be revised from the linguistic point of view by a native translator.
- The services of an expert in translation has been required and the manuscript has been proofreading.
7.2. When expressing p-values, avoid indicating .000; <.001 should be used.
- Done
7.3. Keywords must be in alphabetic order.
- Done
7.4. Line 41. The first time the acronym TPSR appears, its meaning must be indicated. It is recommended to include it in line 32 the first time its meaning appears.
- Done
Round 2
Reviewer 1 Report
The authors attempted to address the reviewers' comments. Whilst noting the efforts, the revised version is yet meet the required quality of the journal. The key area of gap is the weak organisation and presentation of information, fidnings, arguments and supporting evidence. In addition, the English writing still hinders greatly the presentation of arguments and the overall quality of the revised manuscript.
Some examples of the presentation that needs clarifications,
- what does"size effect" refer to?
- what do different p values and eTa shown on the Tables specfically refer to?
- Referring to the conculsion "The use of TPSR with secondary school students for at least 60% of the total teaching 448 time led to general improvements in responsibility, the most self-determined motivation, 449 satisfaction of basic psychological needs, resilience, prosocial behaviour and classroom 450 climate, as well as a positive trend in perceived violence." - pls clarify "at least 60%". The same is shown in the abstract.
Author Response
The authors attempted to address the reviewers' comments. Whilst noting the efforts, the revised version is yet meet the required quality of the journal. The key area of gap is the weak organisation and presentation of information, fidnings, arguments and supporting evidence. In addition, the English writing still hinders greatly the presentation of arguments and the overall quality of the revised manuscript.
We really appreciate the reviewer in encouraging us to improve the manuscript. On the other hand, we are very concerned about the English writing and the quality of the revised manuscript because we have required the services of two experts; one in English language and one in Sport Science topic and unfortunately it seems that it has not had the proper effect (we enclosed the both certificates). Anyway, we do try to answer anyone of your comments and the changes to the manuscript are highlight in yellow.
Some examples of the presentation that needs clarifications,
- what does"size effect" refer to?
We are sorry, this is a mistake, and we have corrected in the whole manuscript.
- what do different p values and eTa shown on the Tables specfically refer to?
P values and eTa on the Tables are detailed in the Note of everyone of them.
- Referring to the conculsion "The use of TPSR with secondary school students for at least 60% of the total teaching 448 time led to general improvements in responsibility, the most self-determined motivation, 449 satisfaction of basic psychological needs, resilience, prosocial behaviour and classroom 450 climate, as well as a positive trend in perceived violence." - pls clarify "at least 60%". The same is shown in the abstract.
Sllight changers have been made to the abstract and conclusion sentences to make the idea more understandable. These changes have been reviewed and approved by the translator and proofreader.
In addition, a new sentence has been added in the Method section, participants subsection, describing what means 60% ((it has been clarified in lines 112-113). Specifically: (of the 30 hours per week of clases that students have in educational centres, the GE-EG group receives a minimum amount of 18 hours of instruction from their teachers implementing the TPSR).
In case, this does not meet your requirements, we would greatly appreciate if you could specify what you are trying to communicate to us.

Reviewer 2 Report
Most of the comments have been addressed in greater or lesser detail. However, a thorough revision by a native speaker of the English language in which this manuscript is written is still required.
Author Response
Most of the comments have been addressed in greater or lesser detail. However, a thorough revision by a native speaker of the English language in which this manuscript is written is still required.
We really appreciate the reviewer in encouraging us to improve the manuscript. On the other hand, we are very concerned about the English writing and the quality of the revised manuscript because we have required the services of two experts; one in English language and one in Sport Science topic and unfortunately it seems that it has not had the proper effect (we enclosed the both certificates). The new changes to the manuscript have been highlighted in yellow.

Reviewer 3 Report
The current version of the article has corrected the vast majority of issues suggested in my first report.
I only think that it is necessary to change two issues that will not significantly affect the content of the article.
1. Although I am not a statistician, I believe that a separate analysis should be performed for each instrument. That is, 7 different analyzes, one for each instrument. This would not substantially change the results, but I think it is more appropriate and of course it is the usual practice in the vast majority of articles.
2. This would facilitate presenting the results of each instrument in a separate table (I mean, 7 different tables, one for each instrument), which in turn would facilitate the interpretation of the tables.
Finally, there are two other merely formal aspects that require minor modifications:
a) According to APA standards, when reporting values ranging between 0 and 1, only the decimal part of the number must be indicated.
That is, it is not 0.005, but .005.
b) There is an error throughout the text with the expression "effect size", which is written in the article as "size effect".
Author Response
We really appreciate the reviewer in encouraging us to improve the manuscript.
The current version of the article has corrected the vast majority of issues suggested in my first report.
I only think that it is necessary to change two issues that will not significantly affect the content of the article.
- Although I am not a statistician, I believe that a separate analysis should be performed for each instrument. That is, 7 different analyzes, one for each instrument. This would not substantially change the results, but I think it is more appropriate and of course it is the usual practice in the vast majority of articles.
- This would facilitate presenting the results of each instrument in a separate table (I mean, 7 different tables, one for each instrument), which in turn would facilitate the interpretation of the tables.
We have split up the last table into 7 different ones as you suggested. Now, it is more understandable to readers. However, we have kept the same statistical analysis because new tests involve new data even though the p-values are the same. In addition, this decision implies modifying the Method and Data Analysis sections where we have justified why to use this test and not others. For example, this is one of the sentences in the Data Analysis section: “Before carrying out a MANOVA, Box’s M test of equality of covariance, Bartlett’s and Levene’s test of homogeneity of variance tests were run to check some of the parametric statistical assumptions”.
Finally, there are two other merely formal aspects that require minor modifications:
- a) According to APA standards, when reporting values ranging between 0 and 1, only the decimal part of the number must be indicated.
That is, it is not 0.005, but .005. - b) There is an error throughout the text with the expression "effect size", which is written in the article as "size effect".
Thank you for these two corrections. We have carried them out in the document and the changes have been highlighted with yellow.